# Trans-2-Hexenal-Based Antifungal Packaging to Extend the Shelf Life of Strawberries

**DOI:** 10.3390/foods10092166

**Published:** 2021-09-13

**Authors:** Raquel Heras-Mozos, Adrián García-Moreno, María Monedero-Prieto, Ana Maria Tone, Laura Higueras, Pilar Hernández-Muñoz, Rafael Gavara

**Affiliations:** 1Packaging Group, Instituto de Agroquímica y Tecnología de Alimentos, CSIC, Av. Agustín Escardino 7, 46980 Paterna, Spain; r.heras@iata.csic.es (R.H.-M.); lauhicon@iata.csic.es (L.H.); phernan@iata.csic.es (P.H.-M.); 2Grupo de Tecnología en Envases y Embalajes, ITENE (Unidad Asociada al CSIC), c/Albert Einstein 1, Parc Tecnologic de Valencia, 46980 Paterna, Spain; adrian.garcia@itene.com (A.G.-M.); maria.monedero@itene.com (M.M.-P.); anamaria.tone@itene.com (A.M.T.)

**Keywords:** trans-2-hexenal, 2-nonanone, active packaging, *Botrytis cinerea*, active cellulose pads

## Abstract

Strawberries are valuable because of their nutritional value, but they are also highly perishable fruits. Fungal decay is the overriding factor that alters the overall quality of fresh strawberries. Because no hygienic treatments to reduce the initial microbial load are feasible, molds develop during postharvest when using conventional packaging. In this study, an antifungal packaging system for strawberries was developed to improve safety and quality. Trans-2-hexenal (HXAL), a natural compound in strawberries, was incorporated into the modified atmosphere packaging (MAP) systems. Zero, 100, and 250 µL of HXAL were included in cellulosic pads and were covered with a polyamide coating to control its release. The pads were placed on the bottom of plastic trays; an amount of250 g of strawberries was added, flow packed in micro-perforated PP bags, and stored at 4 °C for 14 days. Fungal infection was monitored during the storage period, and the optical and textural properties of the strawberries were measured at days 0 and 14. Analysis of the package headspace was conducted to check for the HXAL concentration. HXAL was partially retained in the fruits and was converted into hexyl acetate and 2-hexen-1-ol acetate, but this was only measurably present in the headspace of the active systems. Mold growth was fully inhibited in active packaging although the strawberries were softer and darker than those in the control packages. The active package was not as efficient if the fruits were stored under thermal-abuse conditions (15 and 22 °C).

## 1. Introduction

The color, texture, freshness, and known nutritional value of berries are characteristics that make these fruits very appealing to consumers. However, their postharvest preservation is difficult because berries present a fast metabolism and carry a microbiota that includes spoilage microorganisms that, in some berries such as strawberries, cannot be reduced by external washing because of an extremely sensitive surface. Thus, the shelf life of fresh berries is short, even under refrigeration.

Several strategies to reduce postharvest losses have been attempted: (a) combinations of low temperature and headspace compositions rich in carbon dioxide have proven to reduce dehydration, respiration, and ethylene production rates [1], and (b) treatments with pulsed light or ultrasounds [2], (c) coating the fruit with edible polymers [3,4], or (d) the use of active packaging [5,6] have also been proposed.

Active packaging is a novel food processing technology in which the packaging system is actively involved in the improvement of the quality and safety of packaged goods. Some previous attempts to create active packaging systems applied on berries include the incorporation of diverse antimicrobial substances such as 2-nonanone [7], chitosan [8], bacteriocins [9], organic acids, or metabisulfites [10]. However, none have suitably resolved strawberry postharvest fungal decay because of their low efficiency, the development of off-flavors, or their difficulty in industrial implementation.

Strawberries are a source of bioactive compounds with antioxidant properties [11], but they also release substantial amounts of metabolites, including some with antimicrobial activity against *Botrytis cinerea* (trans-2-hexenal (HXAL), hexanal, 1-hexanol, (E)-2-hexen-1-ol, hexanone, nonanal, (Z)-6-nonenal, (E)-3-nonen-2-one, 2-nonanone, methyl salicylate, and methyl benzoate) [12,13]. HXAL has been reported as an organic compound that berries release as a response and defense against the infection *Botrytis cinerea* [14,15]. Furthermore, C-6 aldehydes, at sub-lethal concentrations, affect the protein expression of the mold [16]. Therefore, this aldehyde has been considered as a potential postharvest treatment to prevent gray mold and specifically for use in antifungal packaging systems. An amount of 2-hexenal was injected in the headspace of bags containing strawberries, reducing the incidence of mold decay. Moreover, encapsulated 2-hexenal has been included in polylactic acid [17] and used as active packaging for strawberries, although above 70% of the agent was lost in the film preparation.

In this study, several methods for HXAL incorporation have been explored for the active packaging of strawberries, including their addition in a polymer matrix, absorption in a cellulosic media, injection in corrugated board, or as addition in cellulosic pads. The most efficient packaging designed that inhibited fungal growth on strawberries was analyzed.

## 2. Materials and Methods

### 2.1. Materials

Fresh strawberries (*Fragaria* × *Ananassa*, Fortuna) at the red-ripe stage were bought in a supermarket within 2 days of being harvested (Mercadona, Valencia, Spain). Damaged, non-uniform, unripe, or overripe fruits were eliminated, and the selected fruits were stored at 4 °C until they were packaged in the active or control packaging systems (within 2 h).

*Botrytis cinerea*, obtained from the Spanish-type culture collection (CECT BC03), was used to conduct the in vitro tests on the antifungal properties of the active compounds.

Polypropylene (PP)/ethylene vinyl alcohol copolymer (EVOH)/PP trays, 168 × 120 × 39 mm (0.57 dm^3^), 800 µm thick (Viduca, Alcoy, Spain), were used to conduct this active packaging study. This high barrier tray avoided permeation through the walls, only allowing gas exchange through their upper opening. PP film that was thirty-µm thick wand that was kindly provided by Envaflex (Utebo, Spain) was used to prepare pillow-type bags for flow packaging. Cellulosic pads were used as a potential carrier of the active compounds (Timbrados Albal, Catarroja, Spain). Folding paperboard with a white mineral coating on the external surface and a gray mechanical fiber layer on the internal surface were used to prepare the boxes (Digital Papel Amoros, Biar, Spain). Polyamide Ultramid 1C (PA) (BASF Española, Barcelona, Spain) was used for the coatings.

HXAL, 2-nonanone (NONE), hexyl acetate (HxAc), 2-hexen-1-ol acetate (HxnAc), ethanol, halloysite, and β-cyclodextrin were obtained from Sigma-Aldrich (Madrid, Spain).

### 2.2. Methods

#### 2.2.1. Active Packaging Systems Preparation

Several methods for the inclusion of the active agent in the packaging system were attempted with different results. Table 1 collects a brief description of the active systems. A deeper description is provided in a Appendix A. The method finally selected for the preparation of the active packaging is described in this section.

HXAL, the selected antimicrobial compound, was included in an adsorbent cellulosic pad. To produce this active device, first, an ethanolic solution with 10% PA (*w*/*v*) was evenly spread onto the surface of one side of the pad by using a 100 µm coating rod (LinLab, Logroño, Spain). This polymer was selected because PA provides a high barrier to organic compounds in dry conditions, obtaining good HXAL retention but a medium to low barrier when wet (presence of food), thus facilitating its release [18]. The coating was dried in a homemade forced-air drying tunnel equipped with a 1000 W IR heat source for 2 min. Once dry, increasing volumes of the active compound (0, 100, and 250 µL) were injected into the cellulose pad and were equally distributed in 10 spots on the other surface and were immediately covered with the same polymeric solution. The final coating thickness was approximately 10 µm. Samples were identified as C (control), 100, and 250, respectively.

Pads with the active compound were placed at the bottom of the trays, and 250 g of strawberries were added (Figure 1). Then, the trays were packed with the PP bag, which had previously been perforated with 15 pores (0.5 mm in diameter) in the surface facing the tray opening. In a previous study (not shown), the minimum number of perforations required to keep an internal atmosphere for the fruits with *p*(O_2_) > 0.2 atm and *p*(CO_2_) < 0.01 atm was calculated. The packages were then stored at 4 °C for 14 days. Samples were periodically monitored in terms of the antimicrobial effectivity of the active devices through mold growth and in terms of the evolution of color and texture of the fruits.

#### 2.2.2. Antifungal Activity of the Agents Determined in In Vitro Tests

The antifungal activity of HXAL and NANE against *B. cinerea* was tested by employing the micro-atmosphere method comprising applying different aliquots of the volatile agent on paper disks adhered to the inner part of the Petri dish lid without making direct contact with the culture medium [19]. The culture medium used for *B. cinerea* was potato dextrose agar (PDA), which had been bought in dehydrated form (Scharlau, Barcelona, Spain). The media were prepared by dissolving the amount of powder as indicated by the manufacturer into distilled water, sterilized in an autoclave at 121 °C for 15 min, cooled down to 50 °C, and poured (15 mL) into sterile Petri dishes (90 mm diameter). When solid, the plates were stored at 4 °C, until use.

*B. cinerea* spores were obtained from the solid cultures in the PDA Petri dishes after they had been incubated for 7 days at 25 °C. The liquid inoculum was obtained by pouring 5 mL of sterile peptone water with Tween 80 (0.05%) and then scraping the surface of fungal culture with a Digralsky handle. A 1.5 mL sample of the fungal suspension was transferred to Eppendorf tubes and was shaken to obtain a homogenous spore suspension. A 1 mL suspension was taken, and serial decimal dilutions were prepared using 0.5 mL of peptone water until a concentration of 10^6^ spores/mL was achieved. The spore count was determined using a microscope (Motic B3 Professional Series, Barcelona, Spain) with the Neubauer chamber method. Samples of 3 µL of the spore suspension were inoculated in three equidistant points over PDA medium. Paper disks of 2.5 cm in diameter were adhered on the middle of the Petri dish lid, and different concentrations of the agents were added to it. The Petri dishes were sealed with Parafilm to reduce the loss of the volatile agents and were incubated at 25 °C. The growth was evaluated by measuring two perpendicular diameters of each colony at 3, 5, and 7 days; on the 7th day, the paper disks were removed to eliminate the agent. The cultures were then incubated again for other 3 days; on the 9th day, there was another evaluation to see which volume was fungicidal, i.e., the fungal death was observed, not only their growth inhibition. Fungal growth data were transformed into percentage inhibition by comparing the colony diameters of the fungi in the antimicrobial test (*Ø_agent_*) with those measured in the control tests (*Ø_control_*), using Equation (1):(1)$$Inhibition\;\left(\%\right)\;=\;100\;-\;\frac{{Ø}_{agent}}{{Ø}_{control}}\;\times \;100$$

#### 2.2.3. Antifungal Activity of the Agents Determined in In Vivo Tests

Agents and active materials were tested against *B. cinerea* in “in vivo” tests using 250 g of strawberries per package. Strawberries are fruits with a quick metabolism and respiration rates between 25 and 50 mL CO_2_/kg·h at 10 °C, or 50 and 100 mL CO_2_/kg·h at 20 °C [20,21]. Thus, common packages (flow pack bags or trays) are macro-perforated to allow for a fast input of oxygen and a fast output of carbon dioxide. However, any beneficial effect of the active agent would have been lost in a packaging system with high ventilation. Accordingly, the strawberries were weighted in trays with the pad and were bagged with a 30 µm-thick PP film with 15 perforations of 0.5 mm in diameter, as stated before. To check the amount of HXAL needed to provide a significant fungal inhibition, various amounts of HXAL (50, 100, 250, 500, 1000 µL) were introduced in a glass vial in the box’s center, and 250 g were deposited around the vial; then, the boxes were bagged with a perforated PP film (headspace volume ca. 250 mL). The packaged strawberries were stored at 4 °C for 15 days, and the microbial growth in the strawberries was monitored. The result was recorded as the percentage of strawberries infected in the package at the storage time.

#### 2.2.4. Physical Properties of Strawberries

The color of the strawberries was determined with a CR-300 MINOLTA colorimeter (Minolta Chroma meter, Minolta Camera Co. Ltd., Osaka, Japan) using a standard light source of D65 and a standard observer of 10°. Measurements were taken at three different points located in the equatorial areas of the strawberry, and the average CIE-Lab color parameters L*, a*, b*, C_ab_*, and h_ab_* were obtained.

The texture of the strawberries was determined using a puncture test with a texturometer TA.XT.plus Texture Analyzer (Stable Micro Systems, Godalming, UK) with a Volodkevich Bite Jaws probe. The test parameters were as follows: rate 1 mm/s and 10 mm of maximum penetration at the equatorial area. Both parameters were measured at day 0 and at the end of the storage period (day 14).

#### 2.2.5. Gas Composition in Package Headspace

The amount of HXAL in the headspace of each package was quantified by gas chromatography. A GC model 7890B (Agilent Technologies, Wilmington, DE, USA) equipped with a flame ionization detector (FID) and a HP-5 capillary column 30 m long, 320 µm diameter, and 0.25 µm thickness (Agilent Technologies, Barcelona, Spain) was used. The injector and detector (FID) temperatures were maintained at 200 °C and 300 °C, respectively. The temperature was raised from 40 to 200 at 10 °C/min, with an initial hold time of 3 min and with a final hold time of 5 min. Samples (1 mL) were taken from the head space packaging and were injected with a 1:5 split ratio. The flow rate of helium as a carrier gas was 15 mL/min. An amount of one milliliter of gas was withdrawn from the package headspace with a gas-tight syringe through a septum adhered to the film. The headspace concentrations were quantified using a calibration curve and were obtained by injection of known amounts of HXAL. Samples were taken each day until day 14. Packaging samples included active packages with strawberries and control active packages with 20 mL of water. The controls simulate the effect of humidity (caused by strawberries) on the package but do not present agent absorption.

Headspace compounds were identified using gas chromatography–mass spectrometry (GC-MS) using a GC model 5890 (Agilent Technologies, Barcelona, Spain) equipped with a MS5972 detector and a HP-5MS capillary column 30 m long, 320 µm diameter, and 0.25 µm thickness. The temperature was increased from 40 to 200 °C, with an initial hold time of 4 min at 40 °C, which was increased at 5 °C/min to 140 °C and at 20 °C/min to 200 °C with a holding time of 2 min. The injector temperature was 230 °C, and the temperature of the transfer line was 260 °C. Mass spectrum was obtained with an ionization energy of 70 eV, and the data were acquired between 29 and 400 uma. Samples (1 mL) were taken from the headspace packaging and were injected in splitless mode. Samples were also injected in the GC-FID to monitor the evolution of the identified compounds in the headspace. Their concentrations were calculated by applying their respective calibration curves, obtained by injection of known amounts of HxAc and HxnAc.

#### 2.2.6. Volatile Compounds Concentration in Fruit

The concentrations of HXAL, HxAc, and HxnAc in the fruits were quantified by GC-FID using the same chromatographic conditions as in Section 2.2.5. Here, a puree was prepared with 100 g of strawberries and 100 mL of distilled water. Diverse amounts of the three compounds were added to this puree, which was then stirred and frozen. On the day of the experiment, the samples were defrosted and kept at 23 °C. A SPME fiber (CAR-PDMS-DVB) (Supelco, Bellefonte, PA, USA) was introduced to the puree and was gently stirred over the course of 25 min to absorb the compounds and immediately desorbed for 5 min in the GC injector. After calibration, the samples stored in the active packages were analyzed with the same procedure.

### 2.3. Statistical Analysis

One-way analyses of variance were conducted. The SPSS computer program (SPSS Inc., Chicago, IL, USA) was used. Differences in the mean values were evaluated by the Tukey b test for a confidence interval of 95%. Data were represented as the average ± standard deviation.

## 3. Results

### 3.1. Antifungal Activity of Agents

The antifungal activity of HXAL and NONE was analyzed following the experimental procedure; Table 2 shows the results expressed as a percentage of inhibition.

Table 2 shows that HXAL was the most active agent against *B. cinerea*, presenting inhibition with only 0.5 µL and being effective in totally inhibiting fungal growth with 1.5 µL and being fungicidal with 2 µL. NONE presented inhibition with 2.5 µL, but inhibition was still incomplete in quantities as large as 10 µL. To check whether HXAL and NONE could have any synergistic activity against the fungi, a mixture 1:1 was used. As seen, an antagonistic effect was observed, as 3 µL NH (1.5 µL HXAL + 1.5 µL NONE) was less effective than 1.5 µL of HXAL. Thus, HXAL was selected as the antifungal agent for the remaining study.

To test the activity of HXAL against *B. cinerea* in the presence of fruits, various amounts of HXAL (50, 100, 250, 500, 1000 µL) were introduced in a glass vial in the box’s center with 250 g of strawberries, as described in Section 2.2.3. No fungal growth was observed in the samples with an amount of at least 100 µL. Since the development of the active package could result in HXAL losses, 250 µL was selected as the starting point for the active system design.

### 3.2. Active Packaging Design

This study aimed to develop an antifungal packaging system to be applied in the commercialization of strawberries. After the previous tests, the selected agent, HXAL, was incorporated in diverse packaging systems to check their antifungal effectiveness and the potential shelf-life extension. HXAL (250 µL) was incorporated on the materials (film, tray, or pad) described in Table 1 and was used to pack 250 g of strawberries, and a qualitative observation of fungal decay was used to select the best design.

In the primary trials, HXAL was incorporated in a polymeric coating applied on the PP film, which was used as a flow pack bag. In a first attempt, HXAL was incorporated in a polyamide (PA) solution, and this film forming solution was used to coat the PP film, but the films were scarcely active in reducing decay because most HXAL evaporated during the coat drying process.

A second attempt was made to incorporate HXAL previously encapsulated in β-CD or nanoclays in the PA solution, but little improvement was observed by either coating the PP film or the paperboard. In all cases, the necessary heating in the drying tunnel, which was to remove solvent from the active coating, resulted in large HXAL losses because of its high volatility (boiling temperature 156 °C).

For the next trial, we directly added HXAL to the surface of the paperboard to absorb the compound in its porous matrix and immediately applied a PA coating to retard the agent release. The results showed that the active system was still inefficient.

Finally, HXAL was incorporated in a cellulosic pad and was coated with PA on both surfaces, as previously described (Section 2.2.1). The deposition of the agent in multiple spots and its impregnation on the cellulosic fibers followed by a rapid barrier coating probably reduced HXAL evaporation and loss. Furthermore, the barrier provided by the dry coating also reduced further losses before its use in food packaging. As PA is highly hydrophilic, in the presence of food products with high water activity, the polymer is plasticized, its barrier decreases, and HXAL can be released. Thus, these active pads provided the most interesting results and therefore were selected for the design of the antifungal packaging system.

### 3.3. Microbial Inhibition

Table 3 shows the microbial growth observed in the strawberries packaged with the active pad containing 100 and 250 µL of HXAL over 14 days of storage. Data are expressed as the percentage of strawberries infected by molds. Mold growth started on the strawberries packaged in the control package (pad without HXAL) after one week of storage. From that point, the infection gradually increased, reaching a contamination level of 76% at the end of storage. This confirms the highly perishable character of strawberries despite refrigeration, as reported elsewhere [22]. A test with 50 µL of HXAL showed no relevant antifungal activity, so it was not continued (data not shown).

Regarding the samples packaged in the active systems, no mold growth was observed throughout the 14 days of storage without differences regarding the volume of active compound. Hence, it was confirmed that the developed active systems could reduce decay in strawberries for at least up to 14 days of storage at refrigeration temperature (4 °C).

Figure 2 includes representative photos of the strawberries to show the difference between samples at the beginning and end of the 14 days of storage and with different treatments. The picture at the top right corner shows mold growth in most fruits. No mold can be observed in the strawberries stored in the packages containing the active pads.

### 3.4. Headspace Composition

The HXAL concentration in the package headspace was quantified in both the control and active packages and with packages containing strawberries or water. These latter assays (samples with water) showed HXAL concentration evolution as a consequence of its release from the package influenced by humidity and its egress outside through the perforations and with no retention by sorption in fruits. Figure 3 shows that the concentration of HXAL was greater in the headspace of the active packages containing water, with values at least 10-fold greater along the storage than in the active packages containing strawberries. Furthermore, the concentration in the package headspace with water increased with the increase of HXAL, especially in the second week of storage. However, no significant differences caused by the amount of HXAL that was added were observed in the concentration of HXAL in packages with strawberries. These results indicate that a important part of the HXAL released from the active pad was absorbed by the fruits. HXAL was not detected in the headspace of the control packages.

During the GC analysis of headspace composition, two relevant peaks, other than HXAL, appeared in the chromatogram of the samples containing strawberries. The analysis of these samples using GC-MS revealed the formation of two by products, 2-hexen-1-ol acetate (HxnAc) and hexyl acetate (HxAc). Both are compounds identified in strawberries [15,23] and are responsible for the scents of strawberries and banana, respectively. Additionally, both are additives approved by the FDA as fragrant and flavoring agents that impart freshness to flavors with fruity nuances [24,25]. These two compounds are naturally present in strawberries when wounded by fungi [23], and were reported by Neri et al. [26] as markers or stimulators of Botrytis germination.

Figure 4 shows the concentrations of HxAc and HxnAc in active packaging headspace along the storage; there were no detectable concentrations in the control package headspace (with water). Previous studies detected these compounds after SPME-GC from blended strawberries [27] and from the analysis of the volatile metabolites generated by strawberries after exposure to aldehydes [15]. In this study, the concentration evolution of 2-hexen-1-ol acetate presented a similar profile to values of HXAL, showing a maximum within the first 2 days of storage and values within the same concentration range. Regarding hexyl acetate, the maximum concentration is delayed to the 5th day, and concentrations were slightly smaller. From these results, we can confirm that HXAL favours the expression of these two fruity aroma compounds in berries.

Because these two compounds are generated by fruits in an increasing rate because of the addition of HXAL, the analysis of their content in the fruit could be relevant. Thus, the concentration evolution of HXAL, HxAc, and HxnAc in the fruit was also analyzed using GC. Figure 5 presents the data obtained for the strawberries stored in the active packages containing 100 µL of HXAL and in the control packages. The concentration of HXAL in fruit was 3-fold greater on day 2 when exposed to the active package but rapidly decreased to values closer to those of the controls on day 4 and after. Likewise, HxAc and HxnAc increased by one order of magnitude in the active packages on day 2 and remained high until day 6. This agrees with the results of Wakai et al., who described the formation of these two acetates and the significant reduction of HXAL after treatment [15]. They also proved that the two acetates were less active against *Botrytis cinerea* than they were against HXAL.

### 3.5. Color and Visual Appearance

Color parameters were obtained for the strawberries at the beginning of the experiment (day 0) and at the end of the storage (day 14). As Table 4 shows, all samples became darker with time (L* decreased), and the redness decreased, as indicated by the reduction of the parameters a* and C_ab_*. Regarding the change of color related to the initial time, the strawberries kept in the active systems presented total color differences (∆E) that were higher than those of the control (only few fruits could be analysed because of mold infection). In all cases, these differences could be perceived by the human eye (∆E > 5) [28].

Figure 2 shows that all of the samples became darker with time, especially strawberries in contact with the active pads. These observations agreed with the color parameters in Table 4. Besides that, visible signs of infection by molds in the control samples were observed (see Figure 2, colour analysis was not conducted on wounded fruits).

### 3.6. Texture

Table 5 shows the values obtained in the puncture test for the strawberries, comparing the initial values (day 0) with the values on day 14. The slope and the maximum force can be related to the deformation of the fruit. In this sense, strawberries in contact with the active pads, especially at higher HXAL concentration, show lower values for both parameters, indicating higher deformability or less firmness.

### 3.7. Fungal Decay at Conditions of Thermal Abuse

After the analysis conducted at 4 °C, it was concluded that active packaging containing 100 µL of HXAL can contribute to the reduction of fungal growth on strawberries although some physical deterioration was also observed. Because strawberries are often shelved in the supermarket at temperatures above the conditions of this test, two experiments were conducted at 15 and 23 °C. The packaging process was identical to that already presented at 4 °C: 250 g of strawberries were packaged in trays containing control or active pads with 100 µL of HXAL and the double PA coating. Figure 6 shows the values obtained at both temperatures.

Figure 6 shows that mold grew quickly on the strawberries packaged in the control trays at both temperatures. This growth started at day 2 at both temperatures but was faster at 23 °C, with mold covering above 50% of the fruits on day 4 at 23 °C and on day 7 at 15 °C. Active packaging shows a significant effect on mold growth. At 23 °C, some infection was already observed at day 2, although this was much lower than in the control trays. On days 3 and 4, infection increased quickly although this was below the decay observed in the control packages. At 15 °C, mold growth also started on day 2 in both the control and active trays. In the control packages, one of every five fruits presented some infection from day 2, increasing until most of the fruits were infected on day 7. However, the active packaging kept infection at very low values during the initial days. Nevertheless, on day 7, half of the fruits presented some infection. Figure 7 shows photos of representative packages similar to those seen for the percentage of infected samples in Figure 6 but with different levels of mold growth. Control samples presented a great amount of aerial mycelium, which spread spores over the rest of the fruits. However, infection was much less developed on fruits in the active packages.

These results proved that active packaging at these temperatures can be useful at reducing infections; however, infections cannot be avoided. At these temperatures, HXAL is released rapidly because of its high volatility, but its concentration rapidly declined by escaping through film perforations, making it less effective than at it is at refrigeration temperatures.

## 4. Conclusions

Two natural organic compounds present in strawberries, HXAL and NONE, were studied as potential antifungal compounds. HXAL is much more active and suitable for active packaging development. In this study, the incorporation of the active compound was attempted by including it in active coatings on films and on paper. Incorporating HXAL in a cellulose pad coated with polyamide on both sides successfully retained the compound, with a delayed release when packaged with the fruit. The exogenous addition of HXAL largely delays fungal decay at 4 °C although it affected the firmness and color and increased the generation of two fruit-aroma compounds, HxAc and HxnAc. The active package also improved stability at higher temperatures, 15 or 22 °C, although mold growth inhibition was not as efficient as it was at 4 °C. Although the development of active packaging always implies an extra cost with respect to conventional packages, the increase in shelf life reduces product losses, making this technology cost effective.

## Figures and Tables

**Figure 1 foods-10-02166-f001:**
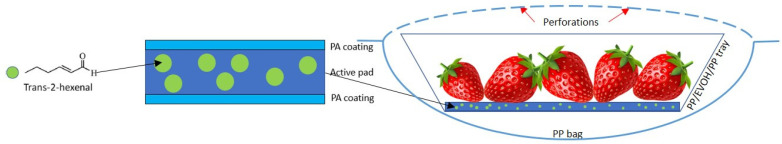
Scheme of the active packaging system developed for strawberries.

**Figure 2 foods-10-02166-f002:**
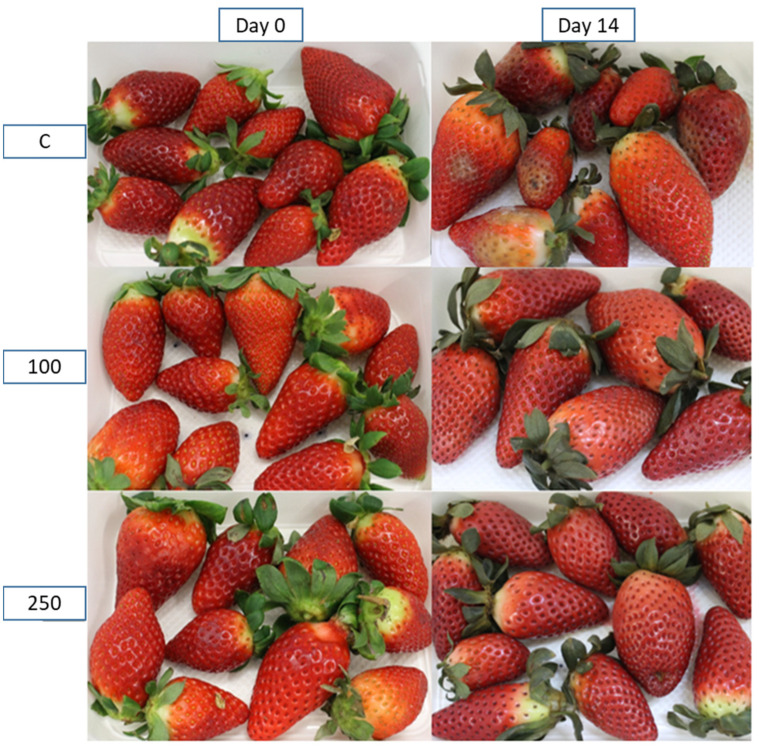
Representative images of strawberries packaged with control and active pads, at the beginning and the end of the storage period. From top to bottom: C, 100 and 250. Left side: day 0. Right side: day 14.

**Figure 3 foods-10-02166-f003:**
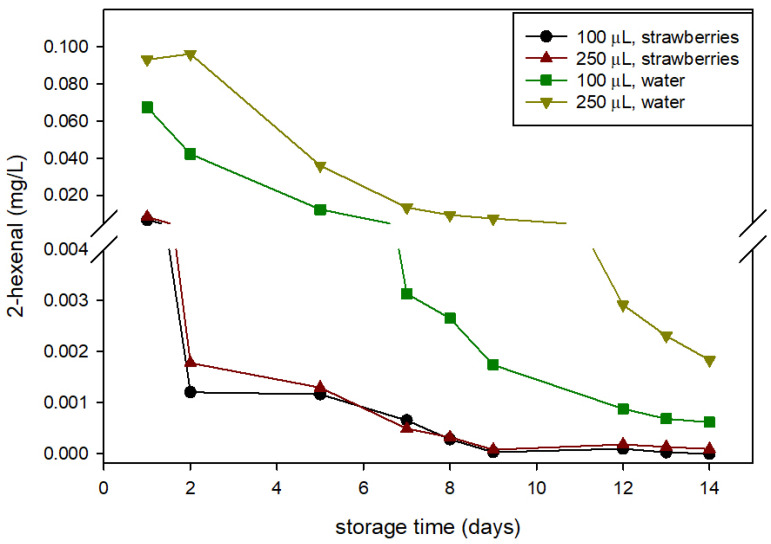
Concentration of trans-2-hexenal (HXAL) in the headspace (HS) of different active packaging with strawberries and with water.

**Figure 4 foods-10-02166-f004:**
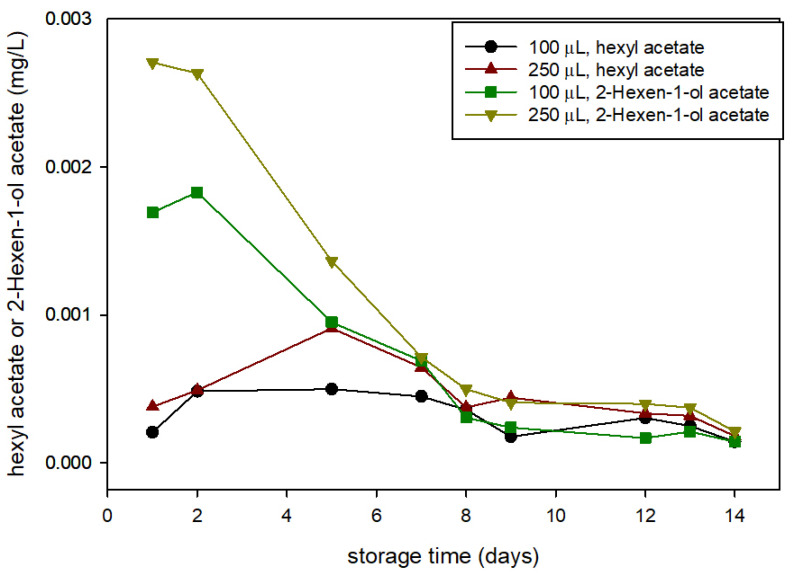
Concentration of hexyl acetate (HxAc) and 2-hexen-1-ol acetate (HxnAc) in head space (HS) of strawberries in active packaging with different quantities of trans-2-hexenal (HXAL) added (100 and 250 µL).

**Figure 5 foods-10-02166-f005:**
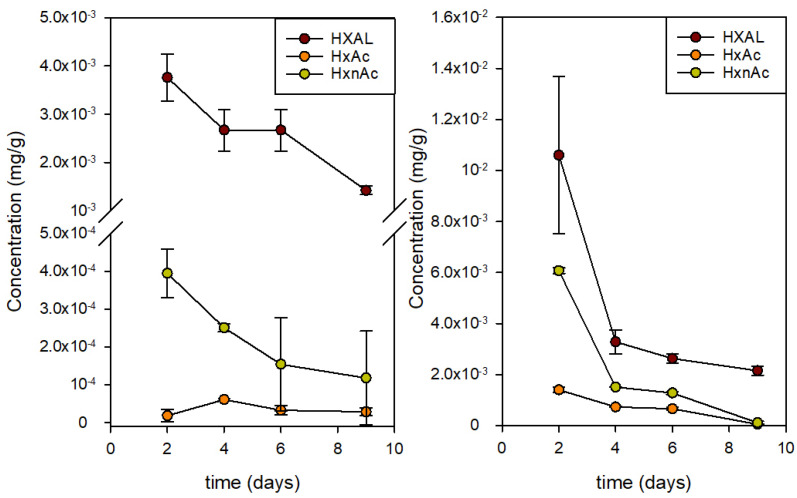
Concentration of trans-2-hexenal (HXAL), hexyl acetate (HxAc), and 2-hexen-1-ol acetate (HxnAc) in strawberries stored in control packages (**left**) and active packages (**right**) with 100 µL of HXAL.

**Figure 6 foods-10-02166-f006:**
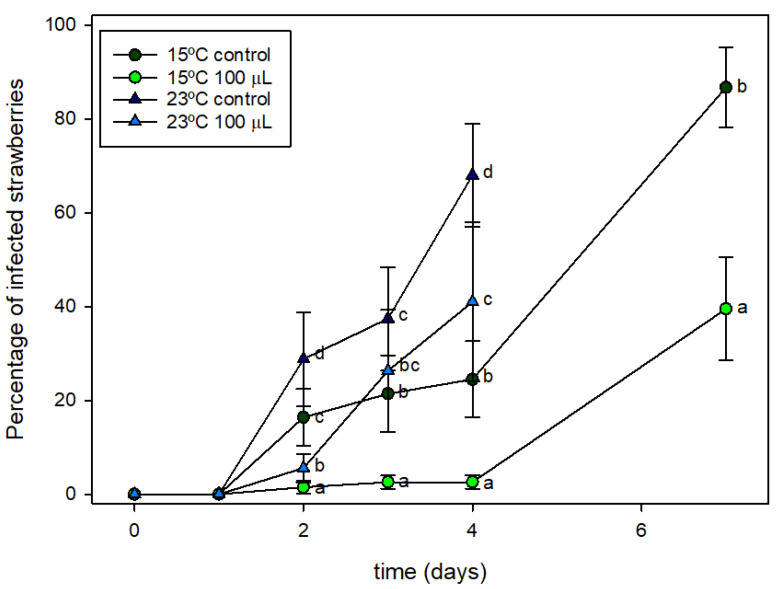
Effect of active component on the fungal decay of strawberries. Percentage of infected fruits per package in the control and active packages (with 100 µL of HXAL) in conditions of thermal abuse (15 °C and 23 °C). ^a,b,c,d^: different letters indicate that there are significant differences between samples or conditions (α < 0.05).

**Figure 7 foods-10-02166-f007:**
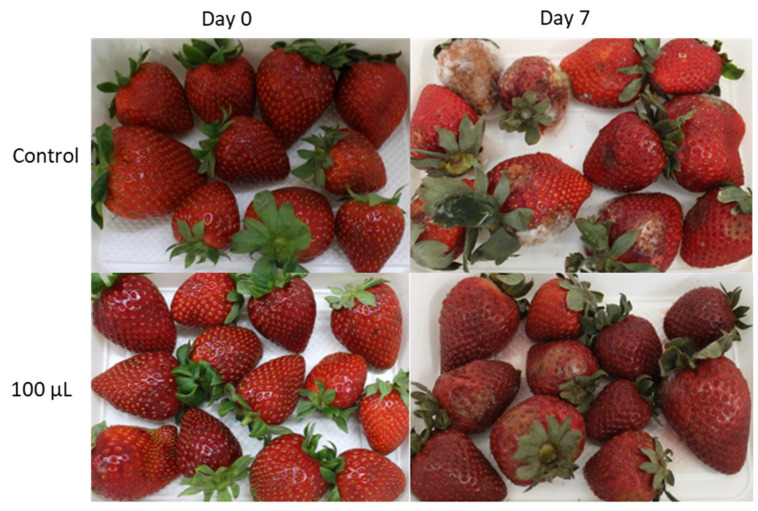
Representative images of strawberry samples in control and active packages (100 µL of HXAL) on day 0 and 7 stored at 15 °C.

**Table 1 foods-10-02166-t001:** Brief description of the diverse methods attempted to efficiently include HXAL in the packaging system. A detailed description is provided in Appendix A.

Method	Method Description	Observations
HXAL in PA coating on PP film	HXAL included in PA ethanolic solution and applied as a coating on PP film (Appendix A)	High loss by evaporation during coating. Low adhesion coating/film.
HXAL encapsulated in CD in PA coating on PP	HXAL encapsulated in β-CD, included in PA solution and applied as a coating on PP film (Appendix A)	Heterogeneous coating, loss by evaporation, loss of transparency, Low adhesion coating/substrate
HXAL-nanoencapsulated in PA coating on paperboard	HXAL and halloysite nanoparticles included in a PA solution and applied as a coating on the unsized surface of paperboard (Appendix A)	Low antifungal efficiency
HXAL injected in paperboard and coated with PA	HXAL distributed by injections on diverse locations of paperboard and coated with PA solution (Appendix A)	Low antifungal efficiency
HXAL injected in cellulosic pad and coated with PA	HXAL distributed and injected on diverse locations of a cellulosic pad and coated with PA on both surfaces (Appendix A)	Good efficiency, low loss of agent

**Table 2 foods-10-02166-t002:** Inhibition percentage (%) of *B. cinerea* exposed to the vapor generated by diverse amounts of trans 2-hexenal (HXAL), 2-nonanone (NONE), and their 1:1 mixture (NH) on days 3, 5, and 7 and after two days of being exposed to air (day 9).

**HXAL (µL)**	**Day 3**	**Day 5**	**Day 7**	**Day 9**
0	0 ^a^	0 ^a^	0 ^a^	0 ^a^
0.5	0 ^a^	11.73 ^b^	9.57 ^b^	3.41 ^b^
1	42.52 ^b^	17.87 ^c^	9.82 ^b^	0 ^a^
1.5	100 ^c^	100 ^d^	100 ^c^	93.90 ^c^
2	100 ^c^	100 ^d^	100 ^c^	100 ^c^
2.5	100 ^c^	100 ^d^	100 ^c^	100 ^c^
**NONE (µL)**	**Day 3**	**Day 5**	**Day 7**	**Day 9**
0	0 ^a^	0 ^a^	0 ^a^	0 ^a^
2.5	70.87 ^b^	13.82 ^b^	3.73 ^b^	2.86 ^b^
5	70.87 ^b^	74.24 ^c^	51.66 ^c^	36.34 ^c^
10	100 ^c^	89.46 ^d^	74.27 ^d^	29.07 ^c^
**NH (µL)**	**Day 3**	**Day 5**	**Day 7**	**Day 9**
0	0.00 ^a^	0.00 ^a^	0.00 ^a^	0.00 ^a^
1	49.61 ^b^	14.52 ^b^	11.41 ^b^	5.07 ^b^
2	64.57 ^c^	25.29 ^c^	13.49 ^b^	15.86 ^c^
2.5	100 ^d^	68.62 ^d^	24.07 ^c^	15.64 ^c^
3	100 ^d^	74.71 ^d^	48.13 ^d^	14.54 ^c^
3.5	100 ^d^	100 ^e^	100^2^	100 ^d^

^a,b,c,d,e^: different letters in a column indicate that there are significant differences between samples (α < 0.05).

**Table 3 foods-10-02166-t003:** Percentage of infected strawberries in the active packaging systems containing 0 (control), 100, and 250 µL of trans-2-hexenal (HXAL) in the cellulosic pad.

Reference	Day 1	Day 5	Day 7	Day 12	Day 13	Day 14
C	0	0	3	17	38	76
100	0	0	0	0	0	0
250	0	0	0	0	0	0

**Table 4 foods-10-02166-t004:** Color parameters of the strawberries on days 0 and 14, L*, a*, b*, C_ab_*, h_ab_*, and ∆E for each reference compared to the initial time.

Day	Sample	L*	a*	b*	C_ab_*	h_ab_*	∆E
0	Control	32.2 ± 0.2 ^a^	33.6 ± 1.6 ^a^	20.0 ± 4.5 ^a^	39.1 ± 3.5 ^a^	30.6 ± 4.7 ^a^	
14	Control	29.5 ± 2.4 ^b^	32.5 ± 4.0 ^a^	17.8 ± 4.4 ^a^	37.1 ± 5.5 ^a^	28.3 ± 3.7 ^a^	3.7
14	100 µL	27.6 ± 3.1 ^b^	29.5 ± 2.4 ^a^	13.5 ± 1.7 ^b^	32.5 ± 2.7 ^b^	24.6 ± 2.0 ^b^	8.9
14	250 µL	27.5 ± 2.4 ^b^	29.3 ± 2.8 ^a^	13.8 ± 2.6 ^b^	32.4 ± 3.5 ^b^	25.1 ± 3.2 ^b^	8.9

^a b^: different letters in a color parameter indicate that there are significant differences between samples or conditions (α < 0.05).

**Table 5 foods-10-02166-t005:** Texture parameters (maximum force (*n*) and slope of the force vs. time curve) of strawberries on days 0 and 14.

Day	Sample	Fmax (*n*)	Slope (*n*/s)
0	Control	5.4 ± 2.4 ^a^	1.2 ± 0.3 ^a^
14	Control	4.6 ± 1.8 ^a^	1.0 ± 0.4 ^a^
14	100 µL	2.5 ± 0.7 ^b^	0.6 ± 0.1 ^b^
14	250 µL	2.7 ± 1.1 ^b^	0.5 ± 0.2 ^b^

^a^^b^: different letters in a colour parameter indicate that there are significant differences between samples or conditions (α < 0.05).

## Data Availability

The data presented in this study are available upon request from the corresponding author.

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
