# Peer review of "Trans-2-Hexenal-Based Antifungal Packaging to Extend the Shelf Life of Strawberries"

_foods, 2021, doi:10.3390/foods10092166_

Round 1
Reviewer 1 Report
The study is well conducted and the manuscript is written clearly. However, brief information about O2 and CO2 in the package headspace would be useful for the readers.
The study evaluated efficacy of anti-fungal Trans 2-hexenal for
controlling fungal infection in packaged strawberries and how to
incorporate it in the packaging system.
It is not completely original as some studies have already reported
ue of this compound. However, this study is slightly different as
it explores different methods of incorporation in packaging system.
Author Response
Thank you very much for your comments. Data about the concentrations of oxygen and carbon dioxide have been included in section 2.2.1. With the perforation of the bag, headspace composition is closed to air.
Reviewer 2 Report
This study looked at the effect of trans-2-hexenal (HXAL) on the changes in quality of packaged strawberries. The overall concept of dosing HXAL in fruit preservation is not new, although the information may be of interest to fruit producers during packaging development to extend product shelf-life. There are several outstanding issues that the authors must address.
- There are considerable grammatical shortcomings; more diligent is needed on the overall grammar and vocabulary; the use of past and present tense should be coherent. The authors should be more careful in proofreading the manuscript following the rules of technical writing e.g., abbreviation should be defined during the first appearance (lines 51), not later (lines 83-84); unit of “days” is “d”; statistical analysis should include the number of determinations for treatment, i.e., the n value; format of reference list is inconsistent with some missing info; etc.
- EVOH/PP (Line 75) multiple layer structure is typically used in packaging applications where oxygen barrier properties are important. Considering that the packaging is perforated, why such structure was used in this study? Also, the dimensions 200 x 155 x 35 (line 75) is confusing; explicitly report what are the dimensions typical SI unit for the packaging industry, e.g., cm or mm. “flow packaging” (line 78) is ambiguous – explain explicitly what the process was to prepare the packaging for testing. Similarly, on line 156, “flow pack bags” is not typical in describing the packaging process.
- Considering that information and experimental conditions are so vague and cannot be duplicated, for brevity, Table 1 should be deleted. Accordingly, information presented in lines 240-257 is not needed.
- The statement “…first, an ethanolic solution with 10% PA (w/v) was evenly spread onto the surface of one side of the pad…” (lines 101-102) should be clearer. I don’t believe the polyamide polymer is soluble in “ethanol”. Please be specific on what was the solvent used and detail the preparation procedure. Also, explain clearly how the spreading process is done, e.g., using wat device? Reporting the volume of the solution deposited on the pad is not adequate; the author must quantify the surface area/thickness and other technical aspects of the absorbent pad, e.g., was it based on cellulosic or nonwoven fibers? what was the supplier?
- The rationale of why 15 pores (line 112) were used in the experiment must be explained in the manuscript. Could this this attribute to the optimal modified atmosphere in the package headspace? Also, Figure 1 – The schematic of the packaging is confusing; it does not depict the relationship of EVOH/PP tray and PP film (both missing) accurately!
- Considering that the authors have access to a gas chromatograph, I am wondering why the release kinetic of HXAL from the carrier was determined in the presence and absence of fruit? This information is essential to establish the release profile of the carrier structure. With 15 holes of 0.5 mm each, the egress of the volatile vapor would have been considerable as well, as stated in lines 156-157. The numbers and units in Line 155 – should be coherent at both temperatures; rephrase.
- The author must report the headspace volume of the package considering that the effective concentration of HXAL in the headspace will determine the efficacy of the treatment. For instance, with a large headspace volume, the same HXAL dosage may not establi the required minimum inhibitory concentration towards the target microorganism.
- Line 160 - The word “diverse” should be replaced with “various”. Be specific, reported what were the amounts used? Also, considering that the release kinetics of HXAL from the vial and the pad carrier are different (due to different exposed area available for diffusion), I am wondering if the information obtained is meaningful? The rationale of “HXAL in glass vial” needs to be better articulated and the rationale should be explained in the manuscript.
- Tables 2 and 3 – Statistical analyses are missing.
- Line 292 – Units for “100 and 250”?
- The statement “…These latter as-says showed the release from the package caused by humidity and its dispersion outside through the perforations, with no retention by sorption in fruits….” Is grammatically flawed and technical not justifiable. Both “humidity” and “sorption” experiments were not conducted in this study. Figure 3 also shows that the presence of fruits resulted in lower HXAL headspace concentration; wasn’t this an indication of vapor absorption by the fruits?
- Line 300 – The rationale of using water in this experiment should be explained; how it is relevant as compared to the actual packaging containing the fruits?
- Lines 303 and 304 – “relevant”???
- Line 306 – What exactly is the “control” package? It is not explicitly explained here nor the method section.
- Figure 3 – The rationale of comparing water and strawberries is not clear; where is the data for the “control” treatment?
- Line 316-317 – grammar; “well additives”??
- Line 319 – incorrect use of word; “bounded”??
- Lines 323 – “solid extraction”??
- Lines 337-338 – Without labelling the exogenous and endogenous HXAL, I am wondering how would one able to determine if the determined HXAL is due to the treatment? The authors should articulate the rationale better here.
- Lines 389 and 396 – Format issue.
- Line 412 – How much less is “much less”?? invoke statistical analysis to indicate if the difference is significant.
- line 433 – The context of “relevant” is ambiguous; be specific, in what way? Also, the conclusion should highlight the issue of substantial firmness loss and discoloration issues, as shown in Tables 4 and 5.
Author Response
Thank you very much for your comments and suggestions. After them, the paper has been greatly improved in this new version. Answers are provided in a file.

Reviewer 3 Report
The manuscript is written properly and the results are explained well in case of antifungal activity, physical properties of strawberries, Gas composition in package headspace data. However, I have a few concerns which need to be addressed.
a) Why did the author choose " Trans-2-hexenal" as packaging material?
b) How feasible and cost effective these packing materials will be on pilot scale or in the packaging industries?
Author Response
Thank you very much for your comments. Our answers are provided after your comments in red.
The manuscript is written properly and the results are explained well in case of antifungal activity, physical properties of strawberries, Gas composition in package headspace data. However, I have a few concerns which need to be addressed.
Information on headspace composition has been added in the revised version (section 2.2.1.)
a) Why did the author choose " Trans-2-hexenal" as packaging material?
As explain in the introduction, we were considering the design of an active packaging system that released a volatile agent effective against fungal decay of strawberries. Essential oils have been reported to be highly efficient from a microbiological point of view but they also damage the sensory quality because of their strong aroma. Thus, we decided to use a compound already present in strawberries. We found 2-nonanone and tras-2-hexenal and we tested them.
b) How feasible and cost effective these packing materials will be on pilot scale or in the packaging industries?
The preparation of PA coatings is a very simple procedure which can be done in industrial scale using coating or printing machinery. We have described this process in previous works (reference 19). The injection of hexenal can be also done in a simple way using a multichannel micropipettes or a multichannel microsyringes. Although any additional process implies an additional cost, the shelf life extension reduces food losses and therefore its worthy to prove it. An additional comment has been added in the conclusion. Thanks.